# Topical Diacerein Decreases Skin and Splenic CD11c^+^ Dendritic Cells in Psoriasis

**DOI:** 10.3390/ijms24054324

**Published:** 2023-02-21

**Authors:** Susanne M. Brunner, Andrea Ramspacher, Caroline Rieser, Julia Leitner, Hannah Heil, Michael Ablinger, Julia Tevini, Monika Wimmer, Andreas Koller, Josefina Piñón Hofbauer, Thomas K. Felder, Johann W. Bauer, Barbara Kofler, Roland Lang, Verena Wally

**Affiliations:** 1Research Program for Receptor Biochemistry and Tumor Metabolism, Department of Pediatrics, University Hospital of the Paracelsus Medical University, 5020 Salzburg, Austria; 2EB House Austria, Research Program for Molecular Therapy of Genodermatoses, Department of Dermatology and Allergology, University Hospital of the Paracelsus Medical University, 5020 Salzburg, Austria; 3Department of Laboratory Medicine, University Hospital of the Paracelsus Medical University, 5020 Salzburg, Austria; 4Research Program for Experimental Dermatology and Glaucoma Research, Department of Ophthalmology and Optometry, University Hospital of the Paracelsus Medical University, 5020 Salzburg, Austria; 5Department of Dermatology and Allergology, University Hospital of the Paracelsus Medical University, 5020 Salzburg, Austria

**Keywords:** diacerein, rhein, psoriasis, imiquimod, dendritic cell

## Abstract

Psoriasis is an inflammatory skin disease characterized by increased neo-vascularization, keratinocyte hyperproliferation, a pro-inflammatory cytokine milieu and immune cell infiltration. Diacerein is an anti-inflammatory drug, modulating immune cell functions, including expression and production of cytokines, in different inflammatory conditions. Therefore, we hypothesized that topical diacerein has beneficial effects on the course of psoriasis. The current study aimed to evaluate the effect of topical diacerein on imiquimod (IMQ)-induced psoriasis in C57BL/6 mice. Topical diacerein was observed to be safe without any adverse side effects in healthy or psoriatic animals. Our results demonstrated that diacerein significantly alleviated the psoriasiform-like skin inflammation over a 7-day period. Furthermore, diacerein significantly diminished the psoriasis-associated splenomegaly, indicating a systemic effect of the drug. Remarkably, we observed significantly reduced infiltration of CD11c^+^ dendritic cells (DCs) into the skin and spleen of psoriatic mice with diacerein treatment. As CD11c^+^ DCs play a pivotal role in psoriasis pathology, we consider diacerein to be a promising novel therapeutic candidate for psoriasis.

## 1. Introduction

Psoriasis is a systemic and chronic inflammatory skin disease characterized by increased neovascularization, keratinocyte hyperproliferation, a pro-inflammatory cytokine milieu and immune cell infiltration. It affects 2–4% of the population with a strong negative impact on quality of life [1,2]. This disease is often associated with comorbidities, such as dyslipidemia, diabetes, and obesity with subsequent cardiovascular complications, and with extra-cutaneous involvements such as arthritis [3,4,5]. The pathogenesis of psoriasis is multifaceted and the exact underlying mechanisms remain elusive. However, the interplay between several immune cell types and diverse cytokines was identified as a pivotal factor in the pathology of psoriasis. Especially T cells, dendritic cells (DCs), and the IL-23/IL-17 axis play a prominent role [6]. Despite continuous advances in elucidating psoriasis pathophysiology and its therapy, existing treatment regimens all have distinct limitations [7], underscoring the need for further research aimed at identifying new drug candidates.

The murine model of imiquimod (IMQ)-induced psoriasiform-like skin inflammation is one of the most frequently used preclinical models of psoriasis [8,9]. IMQ activates Toll-like receptor 7 (TLR7) and shows strong upregulation of the NLRP3 inflammasome leading to a prominent skin inflammation. Importantly, it was shown that IMQ-induced psoriasis is critically dependent on the IL-23/IL-17 axis [8].

Diacerein, an anthraquinone derivative from the rhubarb root, is a slow-acting and anti-inflammatory drug. Deacetylation of diacerein gives rise to its active metabolite rhein, an IL-1 converting enzyme inhibitor. Even though diacerein is considered a non-steroidal anti-inflammatory drug (NSAID), its function differs from classical NSAIDs as it acts on the IL-1β-related pathways rather than on cyclooxygenase 2 (COX2) and prostaglandin E2 (PGE2) regulation [10,11]. Currently, diacerein is approved as an oral formulation for the treatment of osteoarthritis, where IL-1β is a key pathologic player. However, due to severe gastrointestinal side effects, the European Medicines Agency (EMA) established new restrictions for the oral formulation in 2014 [11].

Diacerein reduces the production of caspase-1, resulting in decreased IL-1β maturation and secretion and thus lower AP-1 and NFκB activity with concomitant diminished expression of many pro-inflammatory genes, including IL-1β, TNFα, or IL-6 [10,12,13]. Furthermore, diacerein modulated superoxide anion production, chemotaxis, and phagocytic activity of neutrophils, as well as macrophage migration and phagocytosis in rheumatoid conditions [14]. In keratinocytes, diacerein reverted the effects of IL-1α and IL-1β and dampened the expression of numerous IL-1-responsive genes bearing pro-inflammatory functions [15]. Importantly, the potential of diacerein as a topical treatment was recently shown in two clinical trials. Patients suffering from the rare genodermatosis epidermolysis bullosa were treated topically with 1% diacerein cream. In 22 treated patients, blister numbers decreased significantly after 4 weeks of diacerein application and no treatment-related adverse events were reported [16,17]. Recently, diacerein has been investigated in further multicenter trials, for which results are being eagerly awaited (ClinicalTrials.gov identifiers: NCT03154333, NCT03389308, NCT03472287). In addition, metabolization and deacetylation of diacerein within the skin was demonstrated. Low levels of rhein in human serum and urine substantiated the safety of topical diacerein as compared to oral administration [18].

Since diacerein dampens diverse inflammatory processes in different conditions, we hypothesized that diacerein has a beneficial effect on psoriasis. The possibility to circumvent systemic side effects by topical administration renders this small molecule drug a promising candidate for the treatment of psoriasis. Here, we aimed to evaluate the effect of topical diacerein on IMQ-induced psoriasiform-like skin inflammation in C57BL/6 mice. We tested several doses of diacerein for its effect on psoriasis-associated clinical severity, splenomegaly, and infiltration of immune cells into psoriatic skin and spleen.

## 2. Results

### 2.1. Diacerein Alleviates the Clinical Severity of Psoriasiform-like Skin Inflammation

To investigate the therapeutic effect of topical diacerein in psoriasis, we used the model of IMQ-induced psoriasiform-like skin inflammation in C57BL/6N mice. Psoriasis was induced by daily application of IMQ to depilated back skin. Six hours after IMQ application, the skin was topically treated with different doses of diacerein, placebo, or left untreated. Body weight and macroscopic symptoms of skin inflammation were monitored daily.

Compared to non-psoriatic control mice, IMQ treatment resulted in progressive loss of body weight (*p* < 0.05). Importantly, diacerein did not influence the IMQ-induced loss in body weight nor the body weight in healthy animals (Appendix A).

The progression and the clinical severity of psoriasis were monitored daily using the PASI score, assessing erythema, scaling, and thickening individually. Symptoms first appeared in IMQ-treated mice on day 2 and continued to increase, peaking on day 4 and 5 (thickening) or day 6 (erythema and scaling). Scores were elevated in IMQ-treated mice compared to non-psoriatic control mice for at least one but up to five consecutive days (*p* ≤ 0.0440) (Appendix A, Supplementary Appendix A). Accordingly, the clinical severity of the IMQ-induced psoriasis, represented by the cumulative PASI score, continuously increased until days 5 or 6, respectively (Figure 1, Appendix A, Appendix A).

While erythema and thickening scores were not affected by diacerein treatment, the drug reduced psoriasis-associated scaling (two-way ANOVA treatment main effect, *p* = 0.0146). Over the course of the 7-day treatment period, 5% diacerein and placebo application showed a trend towards decreased scaling (*p* = 0.0762 and *p* = 0.0747, respectively), while application of 2.5% and 10% diacerein diminished the scaling scores (*p* = 0.0076 and *p* = 0.0161, respectively) compared to untreated IMQ-induced psoriasis. Multiple comparison tests revealed that each treatment option resulted in decreased scaling scores on days 5 and 6 compared to untreated psoriatic skin (*p* ≤ 0.0319) (Appendix A, Supplementary Appendix A).

Based on the cumulative PASI scores, we observed a more severe psoriasiform-like inflammation on days 4 to 6 compared to non-psoriatic controls (*p* ≤ 0.0411) in all treatment groups. Remarkably, two-way ANOVA analysis of clinical severity scores revealed a significant interaction between time and treatment in the IMQ groups (*p* = 0.0316). Analyzing single days revealed that placebo alleviated the inflammation on day 4 (*p* = 0.0469) and showed a trend towards reduced severity scores on day 5 (*p* = 0.0544) compared to untreated IMQ-induced psoriasis. In addition, 5% diacerein reduced the clinical severity only on day 6 (*p* = 0.0024). Further, 2.5% and 10% diacerein attenuated the clinical severity from days 4–6 [2.5%: *p* = 0.0023/0.0039/0.0007; 10%: *p* = 0.0129/0.0331/0.0425; (day 4/5/6)] compared to untreated psoriatic skin. Importantly, analyzing the whole 7-day treatment period, multiple comparison tests showed that 2.5% diacerein diminished the clinical severity compared to untreated IMQ-induced psoriasis (treatment main effect, *p* = 0.0072) (Figure 1, Appendix A, Appendix A).

### 2.2. Diacerein Ameliorates Psoriasis-Associated Splenomegaly

As psoriasis is characterized by a systemic inflammation [4,19] which is recapitulated by the IMQ model as splenomegaly [8], we analyzed the spleens of mice at the end of the experiment to investigate a possible systemic effect of diacerein. Two-way ANOVA analysis of the spleen weight revealed a significant interaction between disease and treatment on day 7 (*p* = 0.0063). In general, IMQ treatment induced an enlargement of the spleen compared to non-psoriatic animals (IMQ main effect, *p* < 0.0001). Remarkably, multiple comparison analysis showed that 10% diacerein attenuated the IMQ-induced splenomegaly in comparison to untreated (*p* = 0.0006) and placebo-treated (*p* = 0.0005) psoriasis (Figure 2A).

As we observed that the IMQ-induced body weight loss and clinical severity were declining towards day 7, we decided to measure the spleen weight additionally at an earlier day when the inflammation was more pronounced. Thus, the experiment was terminated on day 4. Since on day 7 placebo treatment had no effect on the splenomegaly compared to untreated psoriasis (Figure 2A), we did not include an untreated IMQ group in this experiment. Interestingly, we observed that spleen weights on day 4 were similar to weights on day 7 (*p* > 0.05). Consistently, diacerein dose-dependently attenuated the IMQ-induced splenomegaly on day 4 in comparison to placebo-treated psoriasis (5%: *p* = 0.0246; 10%: *p* = 0.0005) (Figure 2B).

Further support for a systemic effect of diacerein on IMQ-induced psoriasis was provided when we determined the levels of rhein, the active metabolite of diacerein, in the treated area of the dorsal skin and in plasma of experimental animals. While we observed only a minor amount of rhein in the skin, the rhein levels in plasma were pronounced (Appendix A).

### 2.3. Diacerein Decreases CD11c^+^ Dendritic Cells in the Skin and Spleen during IMQ-Induced Psoriasis

As topical diacerein could modulate psoriasis-associated clinical symptoms and splenomegaly, we sought to elucidate which immune cell types were predominantly affected by diacerein. Therefore, we assessed the distribution of diverse immune cell populations in single cell suspensions from treated dorsal skin and from spleen by flow cytometry.

As 10% diacerein was the only dose consistently attenuating clinical and systemic parameters of IMQ-induced psoriasis, we only included this dose in the flow cytometric analysis. Furthermore, in an attempt to increase the therapeutic window of diacerein, we utilized a higher dose of IMQ to induce psoriasis, applying 80 mg Aldara^®^ topically to the dorsal skin of mice for three consecutive days. Following IMQ, mice were treated with 10% diacerein, placebo or left untreated. Skin and spleen samples were collected on day 4.

Interestingly, the higher dose of IMQ (80 mg Aldara^®^) resulted in similar clinical severity scores as compared to the lower dose (62.5 mg Aldara^®^) (*p* > 0.05) (Appendix A). Remarkably, treatment with 10% diacerein resulted in lower PASI scores following 80 mg Aldara^®^ compared to 62.5 mg. On the third day of treatment, the difference reached significance (*p* = 0.0065) (Appendix A). Comparing the groups treated with 80 mg Aldara^®^, we found that 10% diacerein and placebo had similar attenuating effects on the clinical severity as compared to untreated IMQ-induced psoriasis [two-way ANOVA main treatment effect, *p* = 0.0002 (10%) and *p* = 0.0035 (placebo)], with both treatment options diminishing cumulative scores on day 3 [*p* < 0.0001 (10%) and *p* = 0.0006 (placebo)] and day 4 (*p* < 0.0001) (Appendix A). Furthermore, spleen weights were similar following psoriasis induction with 80 mg Aldara^®^ compared to 62.5 mg Aldara^®^ (*p* < 0.05). Consistently, 10% diacerein showed a strong trend to reduce IMQ-induced splenomegaly (*p* = 0.0665) compared to untreated psoriasis (Appendix A).

At day 4, flow cytometry analysis revealed that IMQ did not alter the number of CD45^+^ leukocytes in the spleen (Figure 3A) or skin (Figure 4A) compared to non-psoriatic controls but changed the relative distribution of immune cell subsets in each tissue.

Using a nine-marker flow cytometric panel, we observed that the fraction of F4/80^+^ macrophages in the spleen were unaffected by IMQ (Appendix A). Compared to non-psoriatic controls, IMQ decreased the numbers of CD4^+^ (*p* > 0.0001) and CD8^+^ T cells (*p* < 0.0001), eosinophils (*p* = 0.0076), and red pulp macrophages (RPMs) (*p* = 0.0016) (Appendix A–D, Figure 3B). Furthermore, IMQ increased the splenic fractions of CD11b^-^CD11c^+^ lymphoid DCs (*p* < 0.0001), CD11b^+^ myeloid cells (*p* = 0.0004), MHCII^+^ and MHCII^-^ monocytes/macrophages (*p* < 0.0001), neutrophils (*p* = 0.0002), myeloid CD11c^+^ DCs (*p* < 0.0001), and CD19^+^ B cells (*p* < 0.00019) and showed a trend to increased NK cells (*p* = 0.0506) (Figure 3C, Appendix A–K).

In the skin, the fractions of CD19^+^ B cells and F4/80^+^ macrophages were unaffected by IMQ treatment (Appendix A). Compared to non-psoriatic controls, IMQ reduced the numbers of CD4^+^ (*p* < 0.0001) and CD8^+^ (*p* = 0.0187) T cells and CD11b^-^CD11c^+^ lymphoid DCs (*p* = 0.0140) (Appendix A–E) in the skin. Further, IMQ increased the fractions of eosinophils (*p* = 0.0266), NK cells (*p* = 0.0041), CD11b^+^ myeloid cells (*p* < 0.0001), neutrophils (*p* = 0.0001), monocytes/macrophages (*p* < 0.0001) and showed a trend to increase CD11b^+^CD11c^+^ myeloid DCs in the skin (*p* = 0.0689) (Appendix A,G, Figure 4B–F).

Interestingly, topical treatment of the psoriatic skin with placebo, but not diacerein, significantly reduced the overall fraction of CD45^+^ cells in the spleen and skin compared to untreated psoriasis (*p* = 0.0017 and *p* = 0.0122;) (Figure 3A and Figure 4A). The majority of immune cell types in the spleen and skin were unaffected by either diacerein or placebo treatment (Appendix A). However, 10% diacerein decreased numbers of RPMs (*p* = 0.0210) and lymphoid CD11c^+^ DCs (*p* = 0.0214) in the spleen compared to placebo treatment (Figure 3B,C). In the skin, both diacerein and placebo treatment diminished the fraction of CD11b^+^ myeloid cells (*p* = 0.0006 and *p* = 0.0054, respectively) (Figure 4B), showed a trend to diminish neutrophil cell numbers (*p* = 0.0938 and *p* = 0.0806, respectively) (Figure 4C), and reduced the IMQ-induced increase in MHCII^+^ (*p* = 0.524 and *p* = 0.0490, respectively), as well as MHCII^-^ (*p* = 0 0127 and *p* = 0.0192, respectively) monocytes/macrophages (Figure 4D,E) compared to untreated psoriatic skin. Remarkably, diacerein treatment additionally decreased the number of CD11b^+^CD11c^+^ skin myeloid DCs compared to IMQ alone (Tukey’s test, *p* = 0.0257) or to placebo (unpaired *t*-test, *p* = 0.0113), whereas placebo treatment had no effect on immune cell populations compared to untreated psoriatic skin (Figure 4F).

## 3. Discussion

The present study aimed to evaluate the effect of the anti-inflammatory drug diacerein on psoriasis. We demonstrated that diacerein has a beneficial effect by attenuating the clinical severity, reducing the disease-associated splenomegaly, and decreasing the infiltration of lymphoid CD11c^+^ DCs to the spleen and myeloid CD11c^+^ DCs to the skin in an IMQ-induced psoriasis mouse model.

In the literature, the anti-inflammatory properties of diacerein have been documented in different inflammatory conditions *in vitro* [13,14,15] and *in vivo*, including osteoarthritis [10], cervical hyperkeratosis [20], and epidermolysis bullosa [17,18]. While these observations render diacerein a promising candidate for the treatment of psoriasis, so far, no studies have tested this hypothesis.

Since oral diacerein exhibits low bioavailability [21] and is associated with diarrhea, liver disorder, and urine discoloration [22,23], other application routes are more preferable. Previously, we observed that a topical 1% diacerein formulation in Ultraphil^®^ cream has shown excellent results in a phase 2/3 randomized, placebo-controlled, double-blind clinical trial in epidermolysis bullosa patients without any study-related adverse events [17]. Furthermore, transdermal delivery of diacerein has been shown to be safe in rodents [24]. Consequently, we tested topical diacerein in experimental psoriasis-like inflammation. As rhein, the active metabolite of diacerein, was detected predominantly in plasma compared to skin, our data prove that the active compound crosses the skin barrier without requiring other vehicles, as reported previously [25,26]. The generally higher rhein levels in skin or plasma of IMQ-treated mice in contrast to non-psoriatic controls is likely due to the disruption of the epidermal barrier, which is characteristic of psoriasis [27]. Accordingly, IMQ-treated skin exhibited increased drug permeation and accumulation [28]. Most importantly, none of the diacerein doses tested in the present study caused any adverse or toxicity-related events nor had any impact on body weight of IMQ-treated or non-psoriatic mice. Of interest, the 10% diacerein dose is >200 times the dose applied to epidermolysis bullosa patients [17], considering differences in body surface area (BSA) and calculating the dose conversion between mouse and human according to Nair et al. [29]. Since we previously found that only 37% of totally applied rhein was detected in a porcine skin model at 72 h after diacerein application [18], we aimed to increase the amount of the active metabolite in the skin. Furthermore, extrapolation of rhein levels in serum from patients who received treatment on 3% BSA with 1% diacerein, to a treatment of 90% BSA, resulted in levels that were still 150-fold below the reported levels upon oral administration [21]. This emphasizes the safety of topical diacerein, even if applied at relatively high doses.

Following psoriasis induction with IMQ, diacerein was applied with a delay of 6 h between topical treatments to avoid direct cross-reactions of the two drugs. In this treatment regimen, diacerein does not prevent or delay disease onset. Instead, diacerein-treated psoriasis exhibits a similar course of the disease compared to untreated psoriasis with clinical severity scores peaking at day 6 and declining thereafter. However, we observed that any kind of topical treatment, i.e., verum or placebo, relieves clinical symptoms compared to leaving the psoriatic skin untreated. This is not unexpected, as Ultraphil^®^ is commonly used as indifferent therapy of subacute and chronic dermatoses. Nevertheless, diacerein had some advantages over placebo, as 2.5% diacerein was the only treatment option showing a significant reducing effect on the clinical severity over the whole 7-day period. This is in line with the reported anti-inflammatory effect of diacerein on human keratinocytes [15]. Diacerein diminished IL-1α/β-induced gene expression of several cytokines and chemokines, which play an important role in psoriasis [15,30,31,32].

While 2.5% diacerein alleviates clinical symptoms, it does not seem to be sufficient for systemic effects, as only the higher doses of diacerein, i.e., 5% and 10%, reduce the IMQ-induced splenomegaly. A possible explanation for the more potent systemic over local effect of diacerein is that rhein levels are high in plasma but low in the skin. It can be speculated that rhein levels in the skin may be too low to exert even more substantial effects on psoriasis severity. A systemic effect of topical diacerein is especially important for the clinical setting as there is increasing awareness that psoriasis is a systemic inflammatory disease [4]. Correspondingly, in patients, psoriasis is not only correlated with spleen enlargement [33,34] but also with splenic inflammation, which is further linked to cardiovascular comorbidities [35]. Interestingly, other inflammatory diseases, such as arthritis, seem to share similar underlying systemic pathological mechanisms with psoriasis [4]. Consequently, arthritis is one of the most common extra-cutaneous involvements in psoriasis, with up to 30% of patients developing psoriatic arthritis [5,36]. Since oral diacerein is approved as an anti-inflammatory treatment for osteoarthritis [10], our data promote a possible effect of topical diacerein on psoriatic arthritis via systemic pathways. However, this hypothesis needs to be tested in future studies.

Since diacerein only showed a slight advantage over placebo treatment regarding the PASI score, we intended to induce a more severe psoriasis by using a higher IMQ dose. Thereby, the placebo effect should be reduced and the therapeutical window of diacerein should be increased to be able to better determine which immune cell populations are affected by diacerein. While 62.5 mg Aldara^®^, representing 3.125 mg IMQ, is the most frequently used dose in the literature to induce psoriasis in mice, the highest dose tested in experimental studies was 80 mg Aldara^®^ (4.0 mg IMQ) [37]. However, we found that this dose failed to increase the PASI score compared to 62.5 mg Aldara^®^. While Baek et al. rated the higher IMQ dose as generating a maximal effect in female BALB/c mice, they failed to show corresponding data [37]. Furthermore, in the initial description of the model, van der Fits et al. reported that the 62.5 mg dose was “empirically determined to cause most optimal and reproducible skin inflammation” in BALB/c and C57BL/6, without showing supporting data as to which doses were tested or which sex of animals was used [8]. As IMQ was demonstrated to have sex- and strain-dependent effects in mice [38], it is possible that a 4.0 mg IMQ daily does not reliably increase PASI scores in male C57BL/6N mice.

To elucidate a possible mechanism of action of topical diacerein, we determined which immune cell types in the skin and spleen are affected. Psoriasis induction with IMQ strongly affects the infiltration of diverse immune cell types to skin and spleen compared to non-psoriatic animals. The IMQ-induced changes to the relative distribution of immune cells observed in the present study are largely in agreement with other previously published reports [8,39,40,41]. Deviations might be due to differences in mouse strains, markers used for the identification of cell types, antibodies used for detection, gating strategies, treatment periods with IMQ, and time points of analysis or, regarding the skin, different isolation protocols.

In the psoriatic skin, placebo and diacerein reduced infiltration of myeloid cells, including monocytes/macrophages and neutrophils, explaining the alleviating effect on the PASI score. Both cell types are reported to be involved in psoriasis pathogenesis. During psoriasis, neutrophils were shown to influence the growth and differentiation of epidermal keratinocytes and to recruit activated effector T cells [42]. Importantly, neutrophils found in psoriatic skin of patients produced IL-17 [43], which seems of critical importance in psoriasis [30,44]. Moreover, macrophages were shown to play a role not only in T cell dependent but also independent psoriasiform inflammation [45]. Macrophages isolated from psoriatic skin of human donors increasingly expressed IL-23 [32], which is important for the polarization of IL-17-producing Th17 cells and thus is another pivotal cytokine in psoriasis pathogenesis [30].

In spleens of psoriatic mice, diacerein reduced the number of RPMs. This splenic cell type is one of the most studied lineages of tissue-resident macrophages, however, evidence for their involvement in immunological functions is scarce. So far, RPMs were implicated in differentiation of T cells towards Tregs and were shown to produce type I interferons (IFN) [46]. Of note, plasmacytoid DCs (pDCs) are potent producers of type I IFN and were identified also as initiators of psoriatic inflammation. Psoriasis-relevant IFNα production by pDCs is induced via TLR9 activation [47]. As RPMs also express TLR9, they might contribute to the systemic inflammation associated with psoriasis [46].

Remarkably, diacerein reduced the infiltration of CD11b^+^CD11c^+^ myeloid DCs in the skin and of splenic CD11b^-^CD11c^+^ lymphoid DCs in psoriatic mice. The important contribution of DCs in psoriasis pathogenesis is emphasized by the observation of highly increased numbers of myeloid CD11c^+^ DCs in the dermis of psoriasis patients. Furthermore, in agreement with our data, disease improvement correlated with reduced numbers of CD11c^+^ cells [48]. Myeloid DCs drive psoriasis progression by stimulating and activating autologous T cells in situ [47]. Therefore, our data indicate that diacerein is able to dampen psoriasis inflammation by reducing the cellular interactions of antigen-presenting DCs with T cells, thereby reducing T cell activation. Specifically important for T cell activation by DCs is the expression of costimulatory molecules, e.g., CD40. Increased CD40 expression is characteristic of DC activation and maturation [49,50]. Interestingly, diacerein was shown to reduce IL-1β-induced CD40 expression on keratinocytes [15], indicating a possible role of diacerein in inhibiting DC maturation and thus T cell activation. Pivotal for psoriatic inflammation, CD11c^+^ DCs were shown to produce TNFα, IL-6, and IL-23 in lesional psoriatic skin [48,51,52]. Antagonization of TNFα with etanercept attenuated psoriasis severity in a clinical trial [53]. Importantly, IL-6 suppresses Treg differentiation and IL-23 drives the polarization of Th17 cells, thus, both cytokines promote psoriasis progression [51]. Mechanistically, diacerein inhibits the IL-1β system but also IL-1-related pathways at different levels [11], resulting in lower AP-1 and NFκB activity, leading to reduced expression of TNFα and IL-6 [10,12,13]. Consequently, diacerein could lower the release of psoriasis-promoting cytokines by DCs. However, whether diacerein has a direct effect on DC maturation and function should be a focus of future studies.

In conclusion, we demonstrated the therapeutic effect of topical diacerein on IMQ-induced psoriasis-like inflammation. Furthermore, we substantiated the safety of topical diacerein. This route of application not only circumvented systemic side effects but instead revealed beneficial systemic effects on psoriasis. This finding not only promotes topical diacerein as a promising candidate drug for the treatment of psoriasis but also for other systemic inflammatory diseases, such as arthritis. Importantly, our study established a reducing effect of topical diacerein on the infiltration of psoriasis-promoting CD11c^+^ DCs to the spleen and skin. Future studies should focus on the direct effect of diacerein on DCs in more detail.

## 4. Materials and Methods

### 4.1. Experimental Animals

All animal procedures were approved by the local ethical committee of the Land Salzburg according to Austrian legislation on animal experiments (20901-TVG/130/6-2019 and 20901-TVG/130/11-2021) and conducted according to the Directive of the European Parliament and of the Council of 22 September 2010 (2010/63/EU).

Male C57BL/6NCrl mice were purchased from Charles River (Sulzfeld, Germany) at the age of 7 weeks. Experimental animals were housed in groups of 3 mice per cage at the Preclinical Research Unit of the Paracelsus Medical University, Salzburg, Austria, under controlled conditions at a 12 h light/dark cycle (lights on/off 0700/1900 h) with unrestricted access to food and water. Prior to the start of experiments, animals were allowed to acclimatize to their new environment for at least one week.

### 4.2. Induction of Psoriasiform-like Skin Inflammation

As previously described [54], to induce psoriasiform-like skin inflammation, an approx. 8-cm^2^ area (4 cm × 2 cm) of the dorsal skin of mice was shaved and depilated (Veet sensitive; Reckitt Benckiser Austria GmbH, Vienna, Austria). The depilation cream was washed off gently but thoroughly. Starting the next morning (0800 h), the bare area of the back skin was treated with Aldara^®^ cream, containing 5% IMQ (3M Pharmaceuticals, Neuss, Germany). Mice received either a daily topical dose of 62.5 mg Aldara^®^ (3.125 mg IMQ) [8,54] for 3 or 6 consecutive days or a daily topical dose of 80 mg Aldara^®^ (4.0 mg IMQ) [37] for 3 consecutive days. Simultaneously, non-psoriatic control mice were treated with a similar amount of a vehicle cream (Vaseline Lanette cream, Fagron, Barsbüttel, Germany). At least 6 h following IMQ or vehicle treatment (1400 h), animals received 100 mg of Ultraphil^®^ cream (Bayer, Vienna, Austria) containing 2.5%, 5%, or 10% diacerein (Duke Chem SA, Barcelona, Spain), equivalent to 2.5, 5, and 10 mg daily dose of diacerein, respectively. Control animals received a similar amount of plain Ultraphil^®^ cream colorized with 0.005% tartrazine (Placebo) or the skin was left untreated. Topical treatments were performed under light isoflurane (IsoFlo^®^, Abbott, Vienna, Austria) anesthesia (isoflurane:O_2_ 2:2% vol). The operator was blinded regarding the content of the Ultraphil^®^ cream. Every day, before any topical treatments, the body weight of mice was recorded and macroscopic images of the dorsal skin were taken. On day 4 or 7, approximately 24 h after the last treatment, mice were deeply anesthetized with a mixture of ketamine (205 mg/kg), xylazine (53.6 mg/kg), and acepromazine (2.7 mg/kg). Blood was collected via cardiac puncture with a heparin-coated syringe and centrifuged at 2500× *g* for 10 min at 4 °C. Plasma was collected, mixed with a protease inhibitor cocktail (cOmplete^TM^; Roche, Penzberg, Germany), and stored at −80 °C. Mice were euthanized by cervical dislocation and the dorsal skin was removed. From the treated area of the dorsal skin, either 8-mm skin biopsies were taken randomly or the complete area was processed into a single cell suspension for flow cytometry. Biopsies were weighed, immediately snap-frozen, and stored at −80 °C. The spleen was excised, weighed, and processed into a single cell suspension for flow cytometry.

### 4.3. Evaluation of Clinical Severity

The severity of the psoriasiform-like skin inflammation was evaluated using a modified clinical Psoriasis Area and Severity Index (PASI) scoring system as previously described [8,54]. The daily macroscopic images of the back skin were used to visually score erythema (redness), scaling (desquamation), and thickening (skin wave formation due to rete ridges) by giving each parameter an independent score between 0 and 4 (with half-point steps). The cumulative score from 0 to 12 (erythema + scaling + thickening) served as an indicator for clinical severity. The semiquantitative scoring was performed independently by two researchers blinded to information on treatments. The mean of values was then calculated.

### 4.4. Quantification of Rhein in Mouse Skin and Plasma

Per mouse, an 8-mm skin biopsy was homogenized in 1 mL fetal bovine serum (Gibco, Carlsbad, CA, USA) using an Ultra-Turrax (IKA, Staufen, Germany). The lysate was centrifuged, the supernatant was collected and stored at −80 °C. Then, 100 µL of skin lysate and 50 µL of plasma were subjected to liquid chromatography tandem-mass spectrometry (LC-MS/MS) analysis on a TripleQuad 5500+ (Sciex, Darmstadt, Germany) for rhein quantification as previously described [16,18]. Rhein levels in the skin were normalized to the weight of the skin biopsy.

### 4.5. Preparation of Skin and Spleen Cell Suspensions

Skin samples were cut into pieces and incubated in C10 medium (RPMI 1640, 10% fetal bovine serum, 1% Pen/Strep, 1% L-glutamine, 1% non-essential amino acids, 1% HEPES, 1% sodium pyruvate, 0.0012% ß-mercaptoethanol; Sigma-Aldrich, Darmstadt, Germany) supplemented with collagenase XI (1 mg/mL), hyaluronidase (0.25 mg/mL), and DNase (0.05 mg/mL) (Sigma-Aldrich) for 45 min in a shaking water bath at 37 °C. Samples were minced through a 100-µm pore size nylon mesh. Cells were collected, centrifuged (300× *g*, 5 min, 4 °C), and washed in C10 medium and counted in a CASY cell counter (OMNI Life Sciences, Bremen, Germany). Spleen samples were minced through a 100-µm pore size nylon mesh, cells were collected and centrifuged (300× *g*, 5 min, 4 °C). Depending on pellet size, cells were resuspended in 3–6 mL red blood cell lysis buffer (150 mM NH_4_Cl, 10 mM KHCO_3_, 0.1 mM EDTA, pH 7.3) for 5 min on ice. Cells were washed in PBS and counted in a CASY cell counter.

### 4.6. Flow Cytometric Analysis

First, 5 × 10^6^ (skin) and 2 × 10^6^ (spleen) cells, respectively, were stained for the markers summarized in Table 1 to discriminate immune cell populations. Additionally, a fixable viability dye was used to exclude dead cells from analysis (Table 1, Appendix A). Staining was done in PBS, supplemented with 10% lamb serum (Gibco, Carlsbad, CA, USA) to block FcγIII/IIR (CD16/CD32). Cells were stained with an antibody cocktail as listed in Table 1. Data were collected using a BD LSRFortessaTM cell analyzer (BD Biosciences, Mountain View, CA, USA). Unstained and single-stained controls were used to set appropriate PMT voltages and to adjust compensations of non-specific fluorescence signals due to spectral overlap using BD FACSDiva software. Fluorescence Minus One (FMO) controls were used to identify gating boundaries. The gating strategies used for data collection are illustrated in Appendix A [39,41,55,56,57]. Further, 300,000 events per sample were recorded. Data analysis was performed using FlowJo v.10 software (TreeStar; BD Biosciences). In the skin, CD45^+^ leukocytes were less than 20% of all live cells, therefore numbers of all assessed immune cell populations in the skin were calculated in relation to CD45^+^ cells. In contrast, in the spleen, CD45^+^ leukocytes were more than 96% of all live cells, thus the number of all assessed immune cell populations in the spleen were calculated in relation to all live cells.

### 4.7. Statistical Analysis

Statistical analysis was performed using Graph Pad Prism 9.3.1 (GraphPad Software Inc., San Diego, CA, USA). Data on daily body weight and severity scores were analyzed by two-way repeated measures ANOVA and Sidak’s or Tukey’s multiple comparison test, as appropriate. Data on spleen weight and rhein levels in plasma and skin were analyzed by two-way ANOVA and Sidak’s or Tukey’s multiple comparison test, as appropriate. In data sets of the population size of inflammatory cells in skin and spleen, outliers were identified by Grubb’s test and were excluded from the statistical analysis. Then, data were analyzed by unpaired *t*-test (vehicle vs. IMQ alone) and one-way ANOVA with Tukey’s multiple comparison test (IMQ groups only). In addition, *p* values < 0.05 were regarded as statistically significant.

## Figures and Tables

**Figure 1 ijms-24-04324-f001:**
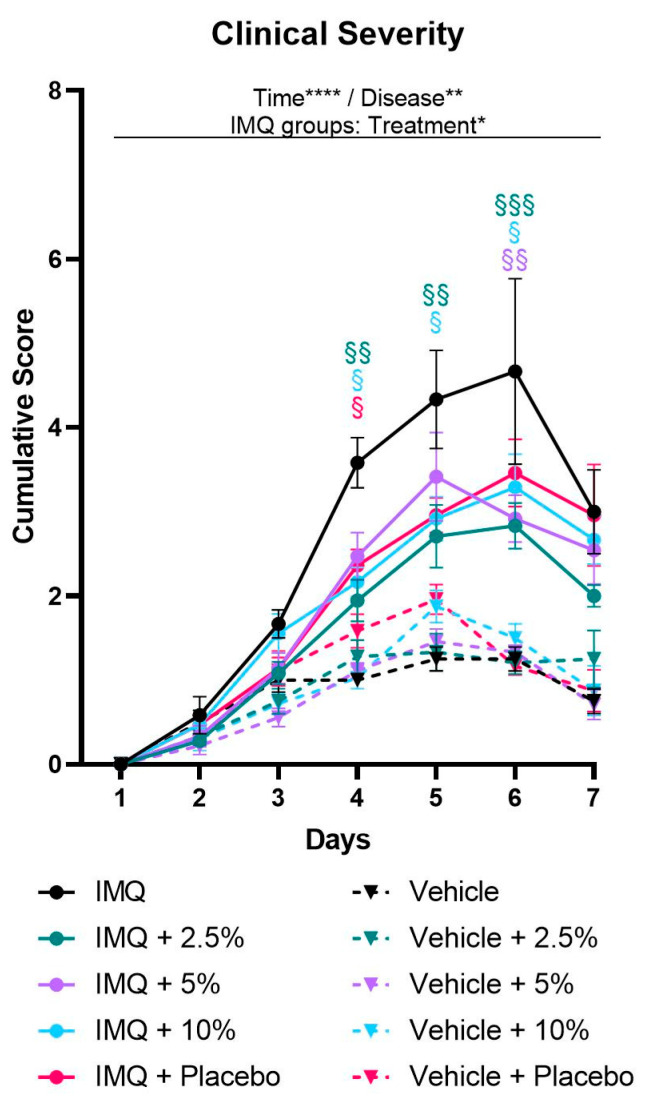
Diacerein ameliorates clinical severity of IMQ-induced psoriasis. Clinical severity is represented by the cumulative score calculated from daily semiquantitative score points for erythema, scaling, and thickening of the dorsal skin. Psoriasis was induced by daily topical application of 62.5 mg Aldara^®^ (IMQ) to the dorsal skin. Following IMQ, mice were treated with 100 mg of 2.5%, 5%, or 10% diacerein or placebo or the skin was left untreated. Control mice received vehicle cream. Data represent means ± SEM. *n* = 3–9 per group. Three independent experiments. * *p* < 0.05, ** *p* < 0.01, **** *p* < 0.0001, repeated measures two-way ANOVA main effects. § *p* < 0.05, §§ *p* < 0.01, §§§ *p* < 0.001 vs. untreated IMQ, Tukey’s multiple comparison test.

**Figure 2 ijms-24-04324-f002:**
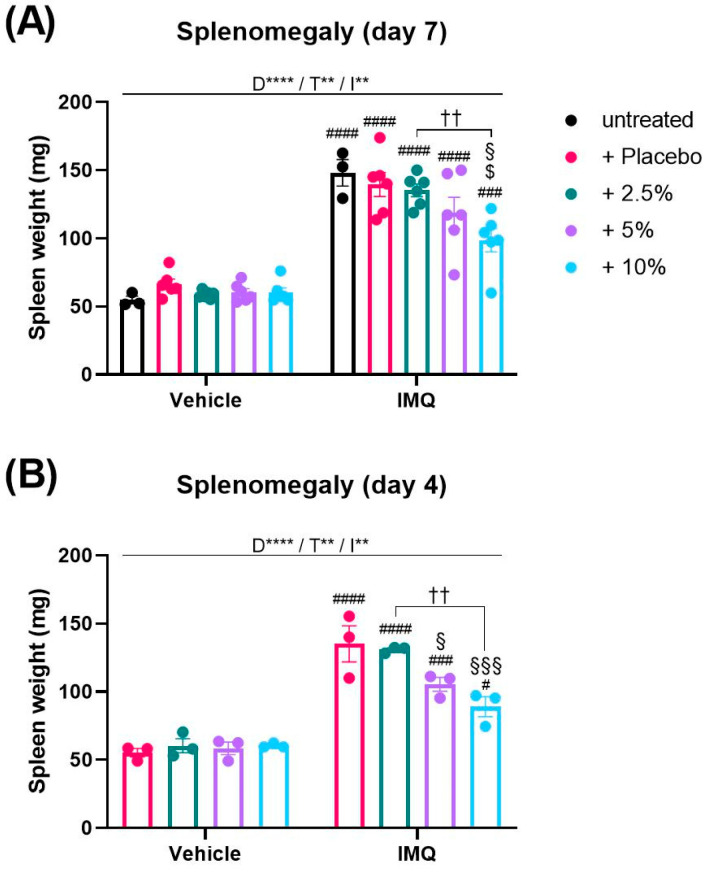
Diacerein reduces IMQ-induced splenomegaly. Spleen weight in mice on day 7 (**A**) after receiving treatment for 6 consecutive days, and on day 4 (**B**) after receiving treatment for 3 consecutive days. Psoriasis was induced by daily topical application of 62.5 mg Aldara^®^ (IMQ) to the dorsal skin. Following IMQ, mice were treated with 100 mg of 2.5%, 5%, or 10% diacerein or placebo or the skin was left untreated. Control mice received vehicle cream. Data represent means ± SEM. *n* = 3–6 per group. Two independent experiments (**A**)., ** *p* < 0.01, **** *p* < 0.0001, two-way ANOVA main effects (D = Disease; T = Treatment; I = Interaction). Sidak’s multiple comparisons test: ^††^
*p* < 0.01; # *p* < 0.05, ### *p* < 0.01, #### *p* < 0.0001 vs. respective vehicle group; § *p* < 0.05, §§§ *p* < 0.001 vs. IMQ + placebo; $ *p* < 0.05 vs. untreated IMQ.

**Figure 3 ijms-24-04324-f003:**
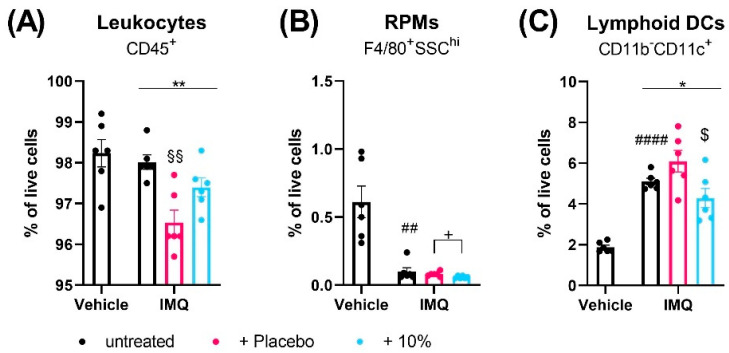
Diacerein affects spleen immune cell populations in IMQ-induced psoriasis. Flow cytometric analysis of CD45^+^ leukocytes (**A**), red pulp macrophages (RPMs) (**B**), and CD11c^+^ lymphoid dendritic cells (DCs) (**C**) in spleen single cell suspensions in mice on day 4 after receiving treatment for 3 consecutive days. Psoriasis was induced by daily topical application of 80.0 mg Aldara^®^ (IMQ) to the dorsal skin. Following IMQ, mice were treated with 100 mg of 10% diacerein or placebo or the skin was left untreated. Control mice received vehicle cream. Data represent means ± SEM. *n* = 5–6 per group. #### *p* < 0.0001, ## *p* < 0.01, unpaired *t*-test, vehicle vs. untreated IMQ; * *p* < 0.05, ** *p* < 0.01, one-way ANOVA, IMQ groups only; Tukey’s multiple comparison test: §§ *p* < 0.01, vs. untreated IMQ; $ *p* < 0.05 vs. placebo; unpaired *t*-test: + *p* < 0.05 as indicated.

**Figure 4 ijms-24-04324-f004:**
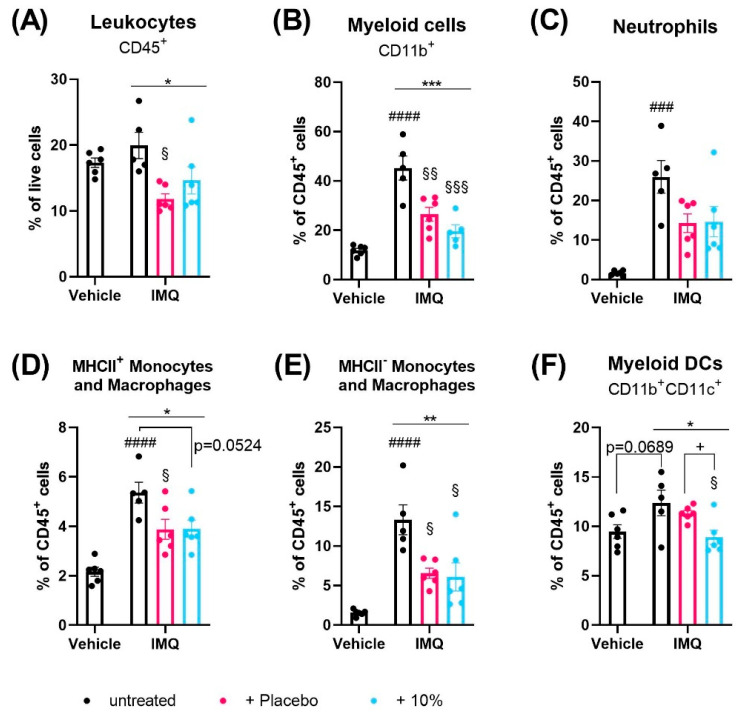
Diacerein affects skin immune cell populations in IMQ-induced psoriasis. Flow cytometric analysis of CD45^+^ leukocytes (**A**), CD11b^+^ myeloid cells (**B**), neutrophils (**C**), MHCII^+^ (**D**) and MHCII^-^ (**E**) monocytes and macrophages, and CD11b^+^CD11c^+^ myeloid dendritic cells (DCs) (**F**) in skin single cell suspensions in mice on day 4 after receiving treatment for 3 consecutive days. Psoriasis was induced by daily topical application of 80.0 mg Aldara^®^ (IMQ) to the dorsal skin. Following IMQ, mice were treated with 100 mg of 10% diacerein or placebo or the skin was left untreated. Control mice received vehicle cream. Data represent means ± SEM. *n* = 5–6 per group. ### *p* < 0.001, #### *p* < 0.0001, unpaired *t*-test, vehicle vs. untreated IMQ; * *p* < 0.05, ** *p* < 0.01, *** *p* < 0.001, one-way ANOVA, IMQ groups only; § *p* < 0.05, §§ *p* < 0.01, §§§ *p* < 0.001, Tukey’s multiple comparison test, vs. untreated IMQ; + *p* < 0.05, unpaired *t*-test: as indicated.

**Table 1 ijms-24-04324-t001:** List of monoclonal antibodies used in flow cytometry analysis of skin and spleen single cell suspensions to detect immune cell populations.

Marker	Clone	Company	CatNo.	Fluorophore	Dilution
NK1.1	PK136	BD Horizon	564144	BUV395	1:100
CD19	6D5	BioLegend	115541	BV650	1:100
CD3ε	145-2C11	BioLegend	100305	FITC	1:100
F4/80	T45-2342	BD Horizon	565612	BV711	1:100
Ly-6G/Ly-6C	RB6-8C5	BioLegend	108407	PE	1:100
CD4	RM4-5	BioLegend	100528	PE/Cy7	1:300
CD8a	53-6.7	eBioscience	17-0081-82	APC	1:400
CD11b	M1/70	Invitrogen	56-0112-82	AF700	1:100
CD11c	N418	eBioscience	48-0114-82	eFluor450	1:100
CD45	30-F11	Invitrogen	45-0451-82	PerCP-Cy5	1:1600
Viability Dye		Invitrogen	65-0865	eFluor780	1:1000

## Data Availability

The data presented in this study are available from the corresponding author upon reasonable request. The data are not publicly available.

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
