# Peer review of "Topical Diacerein Decreases Skin and Splenic CD11c+ Dendritic Cells in Psoriasis"

_ijms, 2023, doi:10.3390/ijms24054324_

Round 1

Reviewer 1 Report

This is an in vivo study performed in imiquimod (IMQ)-induced psoriasis in C57BL/6 mice to test the effect of topical diacerein. Diacerein is a known anti-inflammatory drug modulating IL-1beta pathway, approved for the treatment of ostheoarthritis (OA). In this work, the authors exploited the animal model of psoriasis to test the effect of topical diacerein on clinical severity, splenomegaly and infiltration of immune cells into psoriatic skin and spleen. The article is clear, simple and well structured, and it focuses on a novel topic, therefore it is of interest for a general audience. I have one major comment and few minor comments.

Major comments:

“Methods”: The authors focused on infiltration of immune cells into psoriatic skin and spleen, suggesting that diacerein effectively reduces inflammation in psoriatic mice. This is a novelty. However, they did not provide any result on the effect of diacerein in intercepting IL-1b pathway. Did they perform at least bulk RNA sequencing (or western blot on skin biopsies homogenates) to demonstrate that diacerein is able to intercept this pro-inflammatory pathway?

Minor comments:

“Introduction”: “This disease is often associated with comorbidities, such as arthritis, dyslipidaemia, diabetes and obesity with subsequent cardiovascular complications”. In my opinion, arthritis is not a comorbidity in psoriatic patients. It is more reasonably an extra-cutaneous involvement. I suggest revising this sentence, and similarly one sentence in the “Discussion” chapter.

“Introduction”: “In keratinocytes, diacerein reverted the effects of IL-1α and IL-1β and dampened the expression of numerous IL-1-responsive genes bearing pro-inflammatory functions”. The authors report here a citation that needs to be verified, since it does not focus on keratinocytes (doi:10.1016/j.intimp.2008.01.020).

Figures 3-4-S6-S7: It is not clear to which columns the significance “stars” or “symbols” apply. I suggest homogenizing using the same shape of the lines for all the “p” values shown.

Author Response

Please find a point-by-point response to the comments including detailed descriptions of the revisions made to the manuscript in the attached file "ijms-2122866-Coverletter_Response to Reviewers.pdf".

Thank you.

Reviewer 2 Report

The results presented in this manuscript are interesting but due to the phase of this trial I recommend the following.

1. The aim of the study was to evaluate or assess the effect of topical diacerein on IMQ-indiced psoriasiform-like skin inflammation in C57BL/6 mice, not to evaluate the therapeutic efficacy due to the phase of this trial. Therapeutic efficacy has to be assessed in clinical trials. Please rephrase the aim of the study.

2. To Include photos of the change in modified PASI to let the readers to see the clinical changes at day 7 in all the groups. The tables in the supplementary file is helpful but the photographs could explain more about the differences found in the study.

Author Response

(The authors gave the same response as above.)

Round 2

Reviewer 1 Report

I have no other comment regarding this manuscript. I understand the reasons for not including the additional analyses in the manuscript and the authors' willingness to compile more accurate data on DCs in the future.